# FingER: Fact-Level Answerability for Explainable Refusals in Multi-Hop RAG

## Abstract

Large language models (LLMs) are extensively adopted in retrieval-augmented generation (RAG) systems for solving multi-hop reasoning tasks. While prior works effectively utilize retrieved external knowledge, they often neglect internal factual knowledge in the LLM, resulting in excessive answer refusals with limited explanations. To address this, we propose FingER (Fine-grained Explainable Refusal), a post training approach aimed to elicit the model's ability of using its internal factual knowledge when the external knowledge is missing. Furthermore, FingER is able to provide well-reasoned, explainable justifications for its refusals by analyzing the fact verification status at each step of a multi-hop process. Experimental results on MuSiQue dataset demonstrate that FingER effectively balances accuracy with appropriate abstention, enhancing the reliability and trustworthiness of multi-hop RAG settings.

## 1 Introduction

Large language models (LLMs) have achieved remarkable progress across diverse NLP tasks (Brown et al., 2020; OpenAI, 2023; DeepSeek-AI et al., 2025; Yang et al., 2024; 2025). For knowledge-intensive applications, Retrieval-Augmented Generation (RAG) couples parametric knowledge with a non-parametric memory: a retriever surfaces supporting passages from large corpora, and the generator (LLM) conditions on these passages to produce grounded, context-rich responses (Lewis et al., 2020; Guu et al., 2020; Izacard & Grave, 2021b). Such applications include open-domain question answering (e.g., Natural Questions) (Kwiatkowski et al., 2019), fact verification (e.g., FEVER) (Thorne et al., 2018), and entity linking (Petroni et al., 2021). Beyond single-hop fact finding, many real queries require composing evidence from multiple passages, making multi-hop retrieval and reasoning a central challenge for RAG systems.

Although RAG is able to answer multi-hop questions to a certain extent, it still faces challenge that its external knowledge retrieval process often returns incomplete or partially relevant evidence, leaving one or more necessary premises missing (Sun et al., 2025; Song et al., 2025). Refusal-Aware Instruction Tuning (RAIT) methods have been used to alleviate hallucination brought by incomplete premises (Sun et al., 2025; Song et al., 2025). RAIT is capable of reducing hallucination by enforcing models to express uncertainty when the question is unanswerable.

However, existing RAIT approaches fall short along two dimensions in multi-hop RAG. First, the answerability decision is typically made at the query level and conditioned primarily on the retrieved context, with little consideration of the model's internal factual knowledge at the level of individual premises/hops. As a result, when a prerequisite hop is missing from retrieval but is in fact known by the model, the system is biased toward over-refusal. Second, most RAIT methods supervise refusals through fixed templates (e.g., generic "I don't know" strings or an explicit `[IDK]` token), yielding abstentions that are not localized to the evidence state. In multi-hop RAG, users need hop-localized justifications—which premise is unsupported and why the chain cannot proceed—rather than an undifferentiated refusal. These gaps motivate our fact/hop-level formulation that fuses external evidence with marked internal completions and resorts to abstention only when the fused support remains insufficient.

We introduce FingER, a post-training RAIT method that takes both external knowledge and internal knowledge (at hop-level) into consideration, addressing the fallbacks mentioned above. Specifically, a multi-hop question is considered answerable iff each hop's knowledge is either provided in

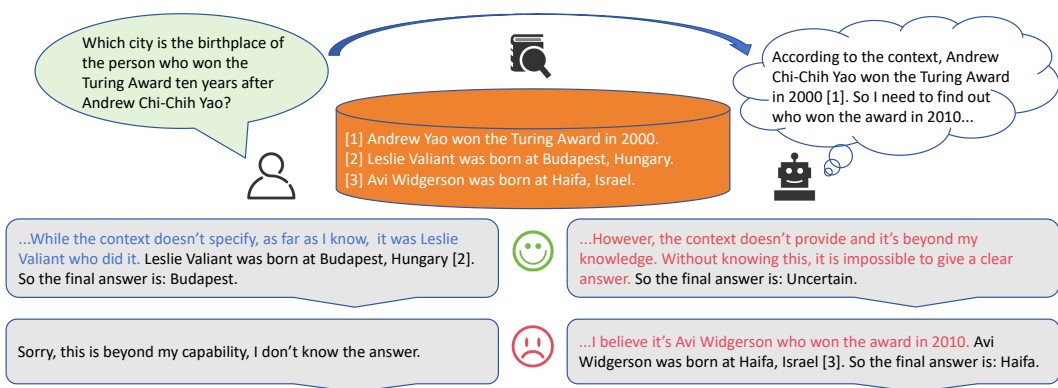

Figure 1: Key idea of FingER. Incomplete evidence is common in multi-hop RAG settings. The answerability should be dependent on whether you know "Leslie Valiant won the Turing Award in 2010" or not. If you do know, supplement the reasoning chain with internal knowledge (up-right), otherwise explain you're unable to answer because of not knowing this (up-left). Hallucinations (bottom-right) and abstention (bottom-left) without explanations are unexpected.

retrieval or acknowledged by the model itself. This goes beyond the previous definition of answerability. What's more, to help models learn to make explainable refusals, we carefully construct a training dataset for DPO. During the reasoning process, when encountering a new hop, the model is expected to first look for retrieved context. If the key fact is missing in the context, i.e. incomplete evidence, the model should utilize its parametric knowledge. If the fact is known, the model should supplement the missing fact and continue the reasoning process. Otherwise the model should clearly express its uncertainty at this step, therefore generating a explainable refusal feedback.

Our contributions can be summarized as follows.

1. We formalize fact-level answerability for multi-hop RAG by jointly assessing internal (parametric) and external (retrieved) coverage per prerequisite fact/hop.

2. We introduce FingER, a post-training RAIT method. FingER constructs type-aware preference data (COMPLETE/SUPPLEMENT/BLOCKED) and trains models to cite, supplement, or abstain with hop-localized explanations.

3. We evaluate our method on benchmark dataset, demonstrating its effectiveness over state-of-the-art methods.

## 2 RELATED WORK

### 2.1 RETRIEVAL-AUGMENTED GENERATION

Retrieval-Augmented Generation (RAG) couples a parametric LM with a non-parametric memory to improve factuality, provenance, and updatability (Lewis et al., 2020; Guu et al., 2020). Early RAG readers such as Fusion-in-Decoder (FiD) aggregate evidence across many retrieved passages (Izacard & Grave, 2021a), while retrieval-pretrained models like Atlas further strengthen few-shot generalization and index refreshability (Izacard et al., 2023). However, multi-hop settings stress RAG because what to retrieve next depends on what has already been inferred. Accordingly, reasoning-augmented pipelines interleave planning and retrieval—e.g., ReAct couples chain-of-thought with tool use (Yao et al., 2023), and IRCoT explicitly alternates CoT steps with retrieval to reduce hallucinations and improve hop-wise grounding (Trivedi et al., 2023). On the evaluation side, multi-hop datasets such as **HotpotQA** (Yang et al., 2018), **2WikiMultihopQA** (Ho et al., 2020), and **MuSiQue** (Trivedi et al., 2022) remain standard testbeds because they supervise supporting facts and connected reasoning. Beyond static "retrieve-$k$" pipelines, recent training recipes aim to make models robust to noisy or incomplete contexts—e.g., Retrieval-Augmented Fine-Tuning (RAFT) teaches models to ignore distractors and cite relevant spans (Zhang et al., 2024b), and Self-RAG learns to retrieve on demand and critique generations within one model (Asai et al., 2024). Yet most

approaches still supervise at the *answer/claim* level and offer limited support for *per-hop complementarity* between external evidence and internal knowledge.

## 2.2 REFUSAL-AWARE INSTRUCTION TUNING

Refusal-aware instruction tuning (RAIT) aims to calibrate models to abstain when knowledge is insufficient. R-Tuning formulates refusal as a meta-skill tied to parametric-knowledge gaps (Zhang et al., 2024a); explicit [IDK] tokens make uncertainty expression more controllable and measurable (Cohen et al., 2024). From a system perspective, Trust-Align integrates grounded attributions with learned refusal for RAG, improving both abstention and citation quality (Song et al., 2025), while the recent DTA framework (Sun et al., 2025) treats honest answering in RAG as a knowledge-boundary problem and aligns models to answer only within the union of retrieved and parametric knowledge. A concurrent survey synthesizes abstention along query, model, and value dimensions and highlights the accuracy–coverage trade-off (Wen et al., 2025). Compared with these lines, FingER conditions refusal on fused internal–external coverage at the hop level, produces localized and auditable refusals (e.g., "missing bridge from A→B"), and separates supplementable cases from truly blocked ones, thereby better matching the practical needs of multi-hop RAG.

## 3 PRELIMINARIES

We denote the query space by $Q$, the corpus by $D$, and the answer space by $A$. A retriever $r : Q \to 2^D$ returns passages $P = r(q)$, and an LLM $M : (x, P) \mapsto a \in A$ generates answers. Let $C(\hat{a}, a^\star) \in \{\text{TRUE}, \text{FALSE}\}$ be a correctness predicate. For a multi-hop query $q$, we assume a canonical decomposition $\text{decomp}(q) = ((s_1, a_1^\star), \ldots, (s_K, a_K^\star))$, where $s_i$ is the $i$-th sub-question and $a_i^\star$ its gold answer.

**Step-level knowledge boundaries.** Parametric step boundary:

$$\mathcal{K}_{\text{param,step}} = \big\{ s \, \big| \, C(M(s, \varnothing), a^\star(s)) = \text{TRUE} \big\}.$$

Retrieval step boundary (with $\text{contains}(\cdot, \cdot)$ meaning a passage contains or directly entails the gold fact):

$$\mathcal{K}_{\text{ret,step}}(q) = \big\{ s \, \big| \, \exists p \in r(q) : \text{contains}\big(p, a^\star(s)\big) \big\}.$$

Fine-grained RAG step boundary:

$$\mathcal{K}_{\text{rag,step}}(q) = \mathcal{K}_{\text{param,step}} \cup \mathcal{K}_{\text{ret,step}}(q).$$

**Reachability and policy.** A step $i$ is *reachable* iff

$$\text{reachable}(i) \iff \forall j < i : \text{SA}(s_j | q) \land C(\hat{a}_j, a_j^\star) = \text{TRUE}.$$

The *should-answer* predicate is

$$\text{SA}(s_i | q) = \big[ \text{reachable}(i) \land s_i \in \mathcal{K}_{\text{rag,step}}(q) \big].$$

Target behavior at step $i$: answer with $\hat{a}_i = M(s_i, P)$ if $\text{SA}(s_i | q)$ holds; otherwise abstain.

**Query-level criterion and certificate.** The query $q$ is answerable iff

$$\forall i \in \{1, \ldots, K\} : \text{SA}(s_i | q).$$

Otherwise abstain at

$$i^\dagger = \min\{ i \mid \neg \text{SA}(s_i | q) \},$$

and emit a certificate $(i^\dagger, \text{reason})$ with reason $\in \{ \neg(s_{i^\dagger} \in \mathcal{K}_{\text{param,step}}), \neg(s_{i^\dagger} \in \mathcal{K}_{\text{ret,step}}(q)) \}$.

## 4 METHODOLOGY

Our objective is to train a model that (i) *answers iff* every reachable sub-question lies within the step-level RAG boundary defined in §3, and (ii) otherwise *abstains* at the earliest failing step with an explicit certificate.

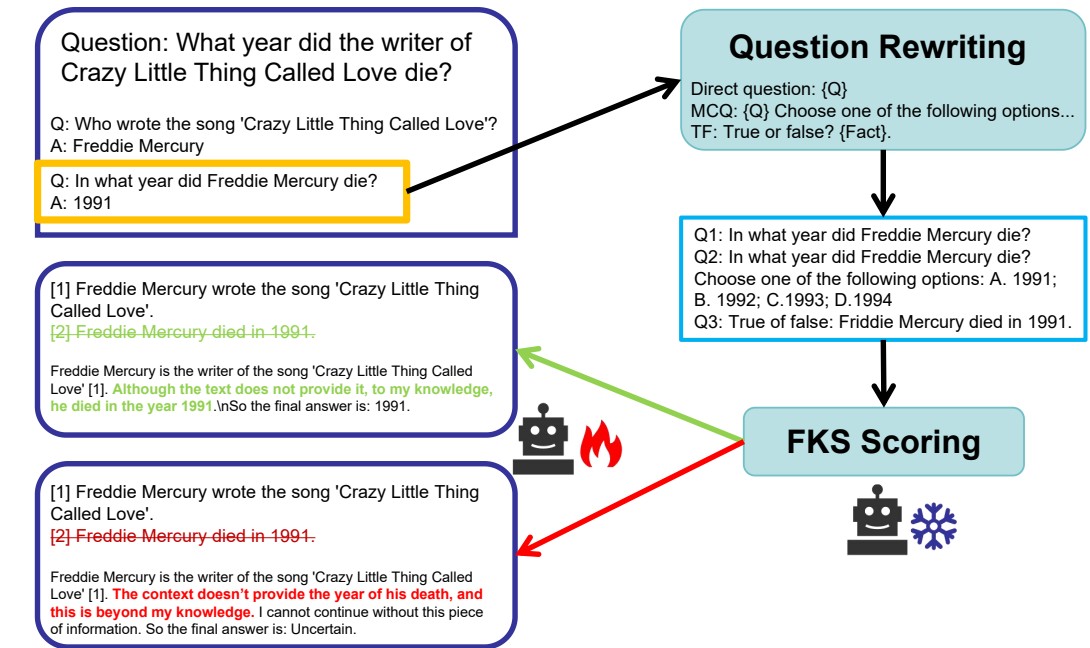

Figure 2: FingER pipeline for constructing fact-level supervision and computing the Fact–Knowledge Score (FKS). (a) From a multi-hop query, we rewrite it into atomic sub-questions. (b) Each sub-question is instantiated with $J$ prompt templates—Direct (Q), Multiple Choice (MCQ), and True/False (TF)—and queried to the LLM. (c) Answers across templates are aggregated into an FKS in $[0, 1]$, estimating whether the model already *knows* the fact from its parametric memory. (d) If $\text{FKS} \geq \tau$ (e.g., 0.8), the fact is labeled *known* and used to continue reasoning (green path); otherwise it is *unknown*, triggering supplementation from retrieved context or an UNCERTAIN decision when evidence is insufficient (red path). The running example shows that when the context omits the year of Freddie Mercury's death, the model can still supply a *known* fact if FKS is high, and abstains when neither context nor parametric knowledge suffices.

## 4.1 DATASET, DECOMPOSITION, AND EVIDENCE PREPARATION

**Dataset and counts.** We use the MuSiQue (Trivedi et al., 2022) training split with its canonical multi-hop decomposition. After filtering a small set of sensitive items from the train split of MuSiQue, we obtain 19,918 multi-hop queries whose decompositions yield 14,582 factual single-hop sub-questions.

**Notation and views to instantiate situations.** For each query $q$, let

$$\text{decomp}(q) = \big((s_1, a_1^\star), \ldots, (s_K, a_K^\star)\big).$$

We construct a *clean full-evidence* context $R^+(q)$ that contains the minimal supporting facts sufficient to answer $q$ end-to-end (no distractors). To control step-wise conditions, we define *single-step masked* contexts

$$R^{-i}(q) = R^+(q) \setminus \{\text{the supporting fact used by } s_i\}, \qquad i = 1, \ldots, K,$$

so masking only affects step $i$ while steps $1{:}i{-}1$ remain reachable under §3.

## 4.2 STEP-LEVEL PARAMETRIC TYPING (FKS PROBE)

To estimate whether a sub-question $s$ is answerable by the model *without* external evidence, we apply a Fact–Knowledge Score (FKS) probe. Let $\{p_j\}_{j=1}^J$ be context-free templates (direct QA / single-choice / verification). From each template we draw $K$ stochastic samples, producing responses $\{r^{(j,k)}\}$. A semantic validator $v(\cdot) \in \{0, 1\}$ checks whether a response entails $a^\star(s)$, and

we define

$$\mathrm{FKS}(s) = \frac{1}{JK} \sum_{j=1}^{J} \sum_{k=1}^{K} v\big(r^{(j,k)}, a^\star(s)\big) \in [0,1].$$

Given a threshold $\tau$, the step-level parametric typing is

$$I(s) = \not\Vdash[\mathrm{FKS}(s) \geq \tau], \qquad \mathcal{S}_{\mathsf{known}} = \{s : I(s) = 1\}, \quad \mathcal{S}_{\mathsf{unknown}} = \{s : I(s) = 0\}.$$

Intuitively, $I(s) = 1$ means $s$ lies in the model's parametric step-level boundary (§3); otherwise it does not.

### 4.3 Response Modes (Targets) and Their Construction

We supervise *targets* as three response modes; inputs are constructed only to instantiate the corresponding situation. Let $\mathrm{SA}(s_i \mid h_i)$ be the should-answer predicate from §3 and $i^\dagger$ the earliest failing step.

**Mode complete (full-evidence answering).**

$$\underbrace{x^{\mathsf{complete}} = (q, R^+(q)),}_{\text{input}} \qquad \underbrace{\mathbf{y}^{\mathsf{complete}}}_{\text{target}} = \text{a step-wise, fully grounded solution using } R^+(q).$$

Construction: produce a faithful chain that cites $R^+(q)$ at each step; since all reachable steps are externally covered, $\mathrm{SA}(s_i \mid h_i) = \mathrm{True}$ for all $i$ and the model should answer rather than abstain.

**Mode supp[$i$] (parametric completion at step $i$).**

$$\underbrace{x^{\mathsf{supp}[i]} = (q, R^{-i}(q)),}_{\text{input}} \quad \text{with } I(s_i) = 1, \qquad \underbrace{\mathbf{y}^{\mathsf{supp}[i]}}_{\text{target}} = \text{answer } q \text{ and } \textit{mark} \text{ step } i \text{ as Parametric.}$$

Construction recipe:

1. Reuse the full-evidence rationale and *truncate* it at the first use of $s_i$; keep steps $1{:}i{-}1$ grounded in $R^{-i}(q)$.
2. At step $i$, insert a declarative marker that the required fact is *absent* from retrieval but available parametrically (since $I(s_i) = 1$), then continue to complete the chain.
3. Output the final answer; no certificate is emitted because $\mathrm{SA}(s_i \mid h_i) = \mathrm{True}$.

**Mode block[$i$] (faithful refusal at step $i$).**

$$\underbrace{x^{\mathsf{block}[i]} = (q, R^{-i}(q)),}_{\text{input}} \quad \text{with } I(s_i) = 0, \qquad \underbrace{\mathbf{y}^{\mathsf{block}[i]}}_{\text{target}} = \textit{refuse} \text{ at } i^\dagger = i \text{ with a certificate.}$$

The construction process is as follows.

1. Produce a faithful partial chain for steps $1{:}i{-}1$ grounded in $R^{-i}(q)$ (these steps remain reachable).
2. At step $i$, assert that the indispensable fact is *absent* from retrieval and $I(s_i) = 0$ (unknown parametrically), so $\mathrm{SA}(s_i \mid h_i) = \mathrm{False}$.
3. Emit the refusal *certificate* $\langle i^\dagger{=}i,\ \mathrm{reason} = \textsc{BothMissing}\rangle$ and stop the chain.

### 4.4 Preference Triples

We construct triples $(x, y^+, y^-)$ so that the preferred response realizes the step-level boundary and the rejected one violates it:

$$(x, y^+, y^-) = \begin{cases} \big(x^{\mathsf{complete}}, \mathbf{y}^{\mathsf{complete}}, \textsc{Short-Refusal}\big), & \text{(unnecessary refusal)} \\ \big(x^{\mathsf{supp}[i]}, \mathbf{y}^{\mathsf{supp}[i]}, \mathbf{y}^{\mathsf{block}[i]}\big), & \text{if } I(s_i) = 1 \\ \big(x^{\mathsf{block}[i]}, \mathbf{y}^{\mathsf{block}[i]}, \textsc{Hallucinated-Completion}(i)\big), & \text{if } I(s_i) = 0. \end{cases}$$

Table 1: Train and Test data statistics

| Model | Train | | | Test | | |
|---|---|---|---|---|---|---|
| | Complete | Supplement | Blocked | Complete | Supplement | Blocked |
| 3B | 5179 | 5821 | 8918 | 805 | 730 | 882 |
| 7B | 5328 | 7205 | 7385 | 822 | 797 | 798 |

## 4.5 TRAINING OBJECTIVE

Let $s_\theta(x, y)$ be the (length-normalized) log-probability under $M_\theta$. For each preference triple $(x, y^+, y^-)$ we optimize Direct Preference Optimization (DPO) (Rafailov et al., 2023):

$$\mathcal{L}_{\text{DPO}}(\theta) = -\log \sigma\big(\beta\left[s_\theta(x, y^+) - s_\theta(x, y^-)\right]\big),$$

and for instruction-style instances $(x, y)$ (all complete and the gold supp/block targets),

$$\mathcal{L}_{\text{SFT}}(\theta) = -\sum_{t=1}^{T} \log p_\theta\big(y_t \mid x, y_{<t}\big).$$

Our final loss mixes them 1:1:

$$\mathcal{L}(\theta) = \tfrac{1}{2}\mathcal{L}_{\text{DPO}}(\theta) + \tfrac{1}{2}\mathcal{L}_{\text{SFT}}(\theta).$$

## 4.6 IMPLEMENTATION NOTES

**Reachability control.** We mask at most one supporting fact at a time to keep steps $1{:}i{-}1$ verifiably reachable; refusals must occur at $i^\dagger = i$.

**Clean context.** $R^+(q)$ contains only gold-supporting facts (no distractors), so supervision targets the interplay of parametric vs. retrieved knowledge rather than distractor robustness.

**Probe settings.** We use $J{=}3$ templates and $K{=}10$ samples for FKS. The thresholds $\tau$ are 0.8 for known boundary and 0.2 for unknown boundary.

## 5 EXPERIMENTS

We design experiments to evaluate our proposed method's ability to generate fine-grained, explainable refusals and its impact on overall performance.

### 5.1 DATASETS

We evaluate FingER on MuSiQue dataset (Trivedi et al., 2022), which contains over 25,000 multi-hop reasoning questions. We perform a simple cleaning step to remove ambiguous or sensitive questions. Following the methodology in Section 3, we generate a train/test set of approximately 19,000 instruction-response pairs from the official train/dev split. Details are shown in Table 1

### 5.2 BASELINES

We compare **FingER** (Fine-grained Explainable Refusal) against the following baselines under the same retriever, context length, and training budget. For a fair comparison, all **DPO-based** methods—**DTA** (Sun et al., 2025), **TrustAlign** (Song et al., 2025), and **FingER**—are trained using the *same pool of positive/negative preference pairs* uniformly sampled from our constructed dataset; only the optimization objectives and supervision granularity differ.

- **Naive (Direct QA).** Directly answers with the base model given the retrieved context; no additional instruction tuning or preference optimization.
- **ICL (Chain-of-Thought)** (Wei et al., 2022). Few-shot in-context exemplars that provide step-by-step rationales; no parameter updates.

- **TrustAlign** (Song et al., 2025). Aligns LLMs for RAG via grounded attributions and learning to refuse; in our setup we adopt a DPO-style preference objective for parity with other DPO baselines.

- **Divide-Then-Align (DTA)** (Sun et al., 2025). Partitions samples by parametric vs. retrieval knowledge boundaries into four quadrants and aligns answering/refusal behaviors accordingly; we implement its alignment as DPO over quadrant-specific preferences for comparability.

### 5.3 EVALUATION METRICS

**From quadrant-level to fact-level answerability.** Unlike the quadrant taxonomy of Sun et al. (2025), which decides answerability by a coarse union of *either* parametric or external knowledge at the query level, our setting instantiates a fact-level boundary: a multi-hop query is answerable iff every indispensable fact $f \in S(q)$ is covered either externally in $D$ or internally by the model. This yields a three-way behavioral taxonomy: COMPLETE (all required facts available), SUPPLEMENT (a missing fact is supplied parametrically), and BLOCKED (some indispensable fact is unavailable both externally and internally).

**Sets and indicators.** For an instance $x = (q, D)$ with gold answer $y^\star$, let $\mathsf{Ans}(x) \in \{0, 1\}$ indicate whether the model outputs a non-abstaining answer and $\mathsf{Abst}(x) = 1 - \mathsf{Ans}(x)$. Let $\mathsf{Corr}(x) \in \{0, 1\}$ mark answer correctness (lexical match with light normalization; details in Appendix). Denote by

$$\mathcal{X}_{\text{in}} = \{x : \mathsf{Ans}(q, D) = 1\} \quad \text{and} \quad \mathcal{X}_{\text{out}} = \{x : \mathsf{Ans}(q, D) = 0\}$$

Note that $\mathcal{X}_{\text{in}}$ subsumes both COMPLETE and SUPPLEMENT cases, and $\mathcal{X}_{\text{out}}$ corresponds to BLOCKED.

**Metric families.** We report the same families as in Sun et al. (2025)—**Overall Quality (OQ)**, **Answer Quality (AQ)**, and **Abstention Quality (AbQ)**—but computed on our fact-level boundary:

**Overall Quality.**

$$\text{OQ Acc} = \frac{\sum_{x \in \mathcal{X}_{\text{in}}} \mathsf{Corr}(x) + \sum_{x \in \mathcal{X}_{\text{out}}} \mathsf{Abst}(x)}{|\mathcal{X}_{\text{in}} \cup \mathcal{X}_{\text{out}}|}.$$

**Answer Quality.** (performance when the instance is *in-boundary*)

$$\text{Rec} = \frac{\sum_{x \in \mathcal{X}_{\text{in}}} \mathsf{Corr}(x)}{|\mathcal{X}_{\text{in}}|}, \quad \text{Prec} = \frac{\sum_{x \in \mathcal{X}_{\text{in}}} \mathsf{Corr}(x)}{\sum_x \mathsf{Ans}(x)}, \quad \text{F1} = \frac{2 \, \text{AQ Prec} \cdot \text{AQ Rec}}{\text{AQ Prec} + \text{AQ Rec}}.$$

**Abstention Quality.** (behavior when the instance is *out-of-boundary*)

$$\text{ARec} = \frac{\sum_{x \in \mathcal{X}_{\text{out}}} \mathsf{Abst}(x)}{|\mathcal{X}_{\text{out}}|}, \quad \text{APrec} = \frac{\sum_{x \in \mathcal{X}_{\text{out}}} \mathsf{Abst}(x)}{\sum_x \mathsf{Abst}(x)}, \quad \text{AF1} = \frac{2 \, \text{APrec} \cdot \text{ARec}}{\text{APrec} + \text{ARec}}.$$

These definitions mirror the OQ/AQ/AbQ intent in Sun et al. (2025) while replacing quadrant membership by our *fact-level* boundary (per-hop complementarity).

### 5.4 MAIN RESULTS

**Overall performance.** Across both capacities, **FingER** achieves the strongest *Overall Quality* while simultaneously improving *Answer Quality* and *Abstention Quality* (Table 4). On 3B, OQ reaches **73.15**% (absolute +8.86 over the best baseline DTA, 64.29%), AQ F1 rises to **68.47**% (+12.35 vs. 56.12%), and AbQ F1 to **82.45**% (+10.08 vs. 72.37%). On 7B, OQ attains **75.09**% (+11.87 over 63.22%), with AQ F1 **72.96**% (+13.81) and AbQ F1 **80.00**% (+12.42). These gains indicate FingER learns a *fact-level* boundary that answers when each hop is supported (externally or parametrically) and abstains otherwise, avoiding the typical over-answer/over-abstain trade-off reported in quadrant-only alignment.

Table 2: Main Results

| Model Name | Method | OQ | AQ | | | AbQ | | |
|---|---|---|---|---|---|---|---|---|
| | | Acc | Rec | Prec | F1 | ARec | APrec | AF1 |
| *Qwen2.5-3B-Instruct* | | | | | | | | |
| naive | | 54.45 | 45.41 | 45.86 | 45.63 | 70.18 | 69.01 | 69.59 |
| ICL | | 52.21 | 38.50 | 44.57 | 41.31 | 76.08 | 61.50 | 68.02 |
| DTA | | 64.29 | 43.91 | 77.74 | 56.12 | **99.77** | 56.77 | 72.37 |
| TrustAlign | | 63.59 | 42.87 | **78.05** | 55.34 | 99.66 | 55.84 | 71.58 |
| FingER-base | | **73.15** | 66.06 | 66.89 | 66.47 | 85.49 | 83.68 | **84.58** |
| FingER-full | | **73.15** | 71.73 | 65.50 | **68.47** | 75.62 | **90.62** | 82.45 |
| *Qwen2.5-7B-Instruct* | | | | | | | | |
| naive | | 60.98 | 55.71 | 61.11 | 58.29 | 71.68 | 60.79 | 65.78 |
| ICL | | 58.96 | 44.84 | 64.65 | 52.95 | 87.59 | 54.02 | 66.83 |
| DTA | | 63.22 | 45.71 | **83.81** | 59.15 | **98.75** | 51.37 | 67.58 |
| TrustAlign | | 62.64 | 44.97 | 82.92 | 58.31 | 98.50 | 51.07 | 67.27 |
| FingER-base | | 73.81 | 71.09 | 70.92 | 71.01 | 79.32 | 79.72 | 79.52 |
| FingER-full | | **75.09** | 75.91 | 70.23 | **72.96** | 73.43 | 87.86 | **80.00** |

**Calibration on the fact-level boundary.** Compared to baselines, FingER shifts the operating point toward *joint* improvements in in-boundary answering and out-of-boundary abstention. Heuristic prompting (`naive`, `ICL`) lacks boundary sensitivity and underperforms on both OQ and AQ F1; TrustAlign and DTA improve abstention but still lag in OQ/AQ under our fact-level metrics (Table 4). Structurally, this mirrors the analysis style in DTA—first report global metrics, then diagnose answer vs. abstain behavior—but FingER replaces query-level quadrants with hop-level complementarity, yielding better calibration under incomplete multi-hop evidence.

**Scaling effects (3B → 7B).** Model scaling modestly lifts OQ by $+1.94$ points ($73.15\% \to 75.09\%$) and AQ F1 by $+4.49$ ($68.47\% \to 72.96\%$), while keeping AbQ F1 high ($82.45\% \to 80.00\%$). Larger capacity thus exploits in-boundary signals better without collapsing abstention quality—consistent with capacity-driven improvements reported in prior boundary-aware alignment studies.

**FingER-base vs. FingER-full.** At 3B, FingER-base and FingER-full tie on OQ (73.15%), but FingER-full nudges the operating point toward higher AQ F1 ($66.47\% \to 68.47\%$) at a minor AbQ F1 cost ($84.58\% \to 82.45\%$). At 7B, FingER-full strictly dominates FingER-base across OQ/AQ/AbQ (OQ $+1.28$, AQ F1 $+1.95$, AbQ F1 $+0.48$), suggesting that mixing SFT with DPO better calibrates answer/abstain decisions as capacity grows.

**Baseline behavior.** `naive` and `ICL` lack explicit boundary supervision and therefore underperform on OQ and AQ F1. TrustAlign and DTA, which incorporate refusal supervision, strike a more balanced behavior but remain below FingER under the same OQ/AQ/AbQ family computed on the *fact-level* boundary (Table 4). This is aligned with observations in DTA that query-level boundary modeling helps, but finer granularity is needed when evidence is missing at specific hops in multi-step reasoning.

## 5.5 ABLATION STUDY

Table 5 quantifies the contribution of each component. Removing **SFT** produces the largest drop (3B: OQ $-12.00$, AQ F1 $-16.16$, AbQ F1 $-4.71$; 7B: $-12.24$, $-12.57$, $-10.95$), indicating instruction-style supervision is essential for realizing fact-level behavior and refusal style. Removing **DPO** yields consistent but smaller declines (3B: OQ $-0.25$, AQ F1 $-1.17$, AbQ F1 $-0.04$; 7B: $-2.23$, $-1.21$, $-4.81$), showing DPO chiefly calibrates decision thresholds rather than teaching core skills. *Pathway* ablations reflect the three-way design: disabling BLOCKED catastrophically collapses AbQ (3B AF1 0.45%, 7B 0.00%) and drags OQ ($-25.90/-21.14$), whereas removing SUPPLEMENT or COMPLETE mainly harms AQ and thus OQ (e.g., 3B AQ F1 $-9.63$ when SUPPLEMENT is removed). This mirrors DTA's practice of dissecting contributions by supervision categories, but

Table 3: Ablation Results

| Model | Method | OQ | AQ | | | AbQ | | |
|---|---|---|---|---|---|---|---|---|
| | | Acc | Rec | Prec | F1 | ARec | APrec | AF1 |
| 3B | FingER-full | **73.15** | 71.73 | 65.50 | **68.47** | 75.62 | 90.62 | **82.45** |
| | w/o SFT | 61.15 | 53.75 | 50.96 | 52.31 | 74.04 | 81.83 | 77.74 |
| | w/o DPO | 72.90 | 66.71 | 67.90 | 67.30 | 83.67 | 81.19 | 82.41 |
| | w/o complete | 70.21 | 67.95 | 62.27 | 64.98 | 74.15 | 88.14 | 80.54 |
| | w/o supplement | 65.25 | 48.34 | **75.18** | 58.84 | **94.67** | 58.39 | 72.23 |
| | w/o blocked | 47.25 | **74.27** | 47.20 | 57.72 | 0.23 | **100.00** | 0.45 |
| 7B | FingER-full | **75.09** | **75.91** | 70.23 | **72.96** | 73.43 | **87.86** | **80.00** |
| | w/o SFT | 62.85 | 64.61 | 56.69 | 60.39 | 59.27 | 82.69 | 69.05 |
| | w/o DPO | 72.86 | 72.64 | 70.89 | 71.75 | 73.31 | 77.18 | 75.19 |
| | w/o complete | 71.53 | 71.22 | 66.23 | 68.63 | 72.18 | 85.21 | 78.15 |
| | w/o supplement | 65.78 | 50.28 | **82.31** | 62.42 | **97.24** | 54.34 | 69.72 |
| | w/o blocked | 53.95 | 80.54 | 53.95 | 64.62 | 0.00 | 0.00 | 0.00 |

at our hop-level granularity it cleanly separates "parametrically completable" from "truly missing" cases—precisely where multi-hop RAG benefits from fact-localized reasoning and refusals.

**Takeaways.** FingER's fact-level boundary yields (i) higher in-boundary precision/recall, and (ii) earlier, localized abstention when a required hop is unsupported by both retrieval and parametric knowledge. Compared to quadrant-level alignment (DTA), this per-hop formulation better matches the failure modes of multi-hop RAG with incomplete evidence, leading to consistent OQ/AQ/AbQ improvements under the same evaluation family.

## 6 CONCLUSION

In this paper, we introduced FingER, a novel framework for generating fine-grained and explainable refusals in RAG systems, particularly for tasks requiring multi-hop reasoning. By training a model to explicitly identify knowledge gaps, assess its internal knowledge, and generate a response that either completes the reasoning or transparently explains the refusal, we move beyond the unhelpful "I don't know" paradigm. Our experiments on MuSiQue (Trivedi et al., 2022) dataset show that this approach not only leads to more trustworthy and helpful models but also improves overall performance by enabling a more synergistic use of retrieved and parametric knowledge. Future work will explore applying this framework to more diverse domains and developing more sophisticated knowledge-probing techniques.

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

# 7 APPENDIX

## 7.1 RESULTS FOR SIMULATED TOP-5 RETRIEVER

Table 4: Main Results

| Model Name | Method | OQ | AQ | | | AbQ | | |
| | | Acc | Rec | Prec | F1 | ARec | APrec | AF1 |
|---|---|---|---|---|---|---|---|---|
| *Qwen2.5-3B-Instruct* | | | | | | | | |
| naive | | 39.39 | 20.98 | 26.79 | 23.53 | 71.43 | 51.85 | 60.09 |
| ICL | | 38.64 | 18.18 | 26.12 | 21.44 | 74.26 | 48.55 | 58.72 |
| DTA | | 37.44 | 4.43 | 20.99 | 7.32 | 94.90 | 39.99 | 56.27 |
| TrustAlign | | 37.57 | 3.13 | **32.65** | 5.71 | **97.51** | 37.89 | 54.57 |
| FingER-base | | 42.37 | 23.13 | 28.31 | 25.46 | 75.85 | 57.52 | 65.43 |
| FingER-full | | **43.32** | **27.88** | 29.83 | **28.82** | 70.18 | **63.03** | **66.42** |
| *Qwen2.5-7B-Instruct* | | | | | | | | |
| naive | | 42.41 | 26.56 | 40.76 | 32.16 | 74.56 | 43.69 | 55.09 |
| ICL | | 40.50 | 21.49 | 36.44 | 27.04 | 79.07 | 43.16 | 55.84 |
| DTA | | 36.91 | 7.97 | 34.04 | 12.91 | 95.61 | 37.44 | 53.81 |
| TrustAlign | | 40.01 | 12.23 | **47.71** | 19.47 | **96.37** | 38.41 | 54.93 |
| FingER-base | | 41.25 | 26.56 | 31.46 | 28.80 | 71.05 | 54.00 | 61.36 |
| FingER-full | | **45.22** | **33.72** | 36.11 | **34.88** | 68.55 | **60.44** | **64.24** |

Table 5: Ablation Results

| Model | Method | OQ | AQ | | | AbQ | | |
|-------|--------|-----|-----|------|-----|------|-------|-----|
| | | Acc | Rec | Prec | F1 | ARec | APrec | AF1 |
| 3B | FingER-full | 43.32 | 27.88 | 29.83 | 28.82 | 70.18 | 63.03 | 66.42 |
| | w/o SFT | 36.62 | 5.34 | 18.85 | 8.32 | 91.04 | 40.51 | 56.08 |
| | w/o DPO | 39.39 | 19.48 | 24.63 | 21.75 | 74.04 | 54.28 | 62.64 |
| | w/o complete | **43.73** | 28.34 | 30.36 | **29.31** | 70.52 | **63.21** | **66.67** |
| | w/o supplement | 38.73 | 4.36 | **44.37** | 7.95 | **98.53** | 38.35 | 55.21 |
| | w/o blocked | 23.05 | **33.36** | 22.07 | 26.56 | 5.10 | 46.39 | 9.19 |
| 7B | FingER-full | **45.22** | 33.72 | 36.11 | **34.88** | 68.55 | **60.44** | **64.24** |
| | w/o SFT | 34.75 | 14.27 | 25.81 | 18.38 | 76.32 | 40.01 | 52.50 |
| | w/o DPO | 40.59 | 33.35 | 32.35 | 32.85 | 55.26 | 58.96 | 57.05 |
| | w/o complete | 45.18 | 31.75 | 37.33 | 34.31 | 72.43 | 55.58 | 62.89 |
| | w/o supplement | 37.24 | 6.86 | **61.67** | 12.34 | **98.87** | 35.27 | 51.99 |
| | w/o blocked | 27.51 | **40.64** | 27.41 | 32.74 | 0.88 | 43.75 | 1.72 |