# OpenReview forum: "FingER: Fact-Level Answerability for Explainable Refusals in Multi-Hop RAG"
_ICLR.cc/2026/Conference — Submitted to ICLR 2026_

### Official Review · Reviewer_QfAe · 2025-10-29

**Soundness:** 3
**Presentation:** 2
**Contribution:** 2
**Rating:** 4
**Confidence:** 3

**Summary:**

This paper mainly proposes to address RAIT issues in multi-hop RAG methods. It focuses on over-refusal triggered by non-explicit information scenarios and multi-stage undifferentiated refusal.

**Strengths:**

1. The motivation of this paper is relatively clear, focusing on the limitations of existing RAIT methods in multi-hop RAGs.

2. The proposed method appears solid.

**Weaknesses:**

1. I think Section 3 needs more detailed explanation. On the one hand, the notation in the paper does not seem to be the conventional representation of RAGs. On the other hand, these representations are not intuitively understandable, so further explanation is recommended.

2. The experiments in the paper are only conducted on the MuSiQue Dataset. It is recommended to extend it to other datasets, such as HotpotQA.

3. The paper does not provide code or other reproduction details.

**Questions:**

See weaknesses above. Please respond to these cons.

---

### Official Review · Reviewer_sinM · 2025-10-30

**Soundness:** 2
**Presentation:** 1
**Contribution:** 2
**Rating:** 2
**Confidence:** 4

**Summary:**

This paper introduces FingER, a fine-grained refusal-aware post-training method for multi-hop retrieval-augmented generation (RAG). Unlike prior work that makes refusal decisions at the query level, FingER leverages both retrieved evidence and internal knowledge at the fact/hop level, enabling more accurate and explainable abstentions. The method constructs step-wise reasoning targets and employs Direct Preference Optimization (DPO) to train the model accordingly.

**Strengths:**

1. The paper explores an interesting idea of combining retrieved context and parametric knowledge at the hop level to make fine-grained answerability decisions.
2. Experimental results on MuSiQue dataset show that the proposed approach outperforms some baselines.

**Weaknesses:**

1. The writing of the paper requires significant improvement. While the work focuses on Refusal-Aware Instruction Tuning, it lacks sufficient background and conceptual explanations to help readers unfamiliar with this area understand its foundations. Moreover, the paper introduces a large number of notations and equations without adequate clarification, making it difficult to follow the technical details. Overall, the paper reads more like a draft than a polished academic submission.
2. The paper lacks comparisons with RAG models designed for multi-hop questions.
3. The paper only conducts experiments on the MuSiQue dataset. It is unclear whether the proposed approach can generalise to other datasets.
4. The analysis of the model’s explainability is insufficient. Although the paper claims that FingER enables "well-reasoned, explainable justifications" for refusals, there is a lack of qualitative or quantitative evidence to support this claim.
5. The code is not provided, raising reproducibiltiy concerns.

**Questions:**

See the Weaknesses section.

---

### Official Review · Reviewer_V1LR · 2025-11-01

**Soundness:** 2
**Presentation:** 3
**Contribution:** 2
**Rating:** 2
**Confidence:** 3

**Summary:**

The paper proposes FingER, a post-training approach for multi-hop RAG systems that aims to leverage both parametric (internal) and retrieved (external) knowledge. The method also trains models to provide explainable refusals by analyzing fact verification at each reasoning step. The authors propose to create a tailored preference dataset for each model, covering the multi-hop answers with know and unknow fact (according to the model), pairing with the decision to answer or abstain the question. The method is trained using a combination of supervised fine-tuning (SFT) and direct preference optimization on the generated preference dataset, and evaluated on the MuSiQue dataset.

**Strengths:**

- The paper addresses a well-motivated problem of combining internal model knowledge with retrieved external information for multi-hop reasoning.
- The idea of issuing refusal certificates provides a step toward interpretable and trustworthy LLM behavior, especially for multi-hop QA where missing evidence can lead to hallucination.
- The authors provide detailed metrics across multiple dimensions (OQ, AQ, AbQ).
- The paper is generally well-structured and easy to follow.

**Weaknesses:**

- The proposed framework relies on several strong assumptions that severely limit its real-world applicability. The method assumes access to a pre-existing decomposition module that can reliably break multi-hop questions into sub-questions with gold answers.  It also appears to assume deterministic or near-deterministic model outputs, but provides no information about temperature settings during generation, how do the framework handle the free-form generation nature of LLMs which may deviate from expected formats.
- The ablation study indicates that SFT on curated data is the dominant contributor to performance improvements. Given that both training and evaluation are conducted on the MuSiQue dataset, the model may simply overfit to the dataset rather than learning the intended skill of integrating internal and external knowledge. All experiments are limited to a single dataset (MuSiQue). Without testing on an unseen dataset, it is impossible to determine whether the proposed method generalizes beyond the training dataset.
- It is not explicitly stated whether the DTA and TrustAlign baselines also underwent SFT before DPO. If not, the performance gains reported for FingER could simply stem from additional fine-tuning rather than the proposed mechanism.

**Questions:**

1. How does the framework operate at inference time?
2. Did the baselines (DTA, TrustAlign) also include an SFT stage? If not, how can we ensure a fair comparison?

---

### Official Review · Reviewer_ornH · 2025-11-01

**Soundness:** 2
**Presentation:** 3
**Contribution:** 3
**Rating:** 4
**Confidence:** 3

**Summary:**

This paper working on improving model's ability of when to refuse answering and provide explainability on refusal during multi-hop RAG. The author claimed that models often fail ungracefully when external (retrieved) knowledge is incomplete. The authors identify two key failings in prior work : (1) systems over-refuse queries, even when the missing information is available in their own parametric (internal) knowledge , and (2) their refusals are often generic (e.g., "I don't know"), failing to explain which piece of information is missing. To address this, the author make the following contribution:

1. Formulate "fact-level answerability": This concept defines a multi-hop question as answerable if and only if every necessary fact (or "hop") in the reasoning chain is supported by either the retrieved external context or the model's internal parametric knowledge.

2. FingER pipeline, that use three distinct modes, COMPLETE, SUPPLEMENT, BLOCKED, to train the model.

Overall, I think the paper's idea is great, but a lot of the experimental setting is unrealistic or unclear to me whether it's practical to real-world multihop RAG. If the author can provide clear explanations, I would consider to raise my evaluation.

**Strengths:**

1. The problem formulation is articulated clearly and precisely. The author distinct between prior "query-level" answerability and the proposed "fact-level" or "hop-level" answerability well.

2.  Thorough Ablation Study: The ablation studies in Table 3 are comprehensive and provide strong evidence for the contribution of each component . The catastrophic drop in AbQ F1 to ~0% when removing the w/o blocked supervision and the large drop in AQ F1 when removing w/o supplement perfectly validate the paper's core design

**Weaknesses:**

Key weakness of the paper is regarding the lack of practiability for the method in realistic situation:
1. Overly "Clean" Evaluation Setting: The main results (Table 2) are generated on a synthetic test set using "clean full-evidence context", without distractor and contexts where only one fact is masked at a time. This is not representative of a real-world RAG setting, which is typically flooded with noisy, irrelevant, and distractor.

2. The method ignores what would be the case when knowledge conflict (between internal knowledge and provided evidence) happens?

3. In real practice, for multi-hop QA, the key is on how to decompose a question. This paper also ignores the details on this point.

**Questions:**

What are the retriever used in the paper?

---

### Meta-Review · Area_Chair_MFL9 · 2026-01-05

**Summary:**

The paper proposes FingER (Fine-grained Explainable Refusal), a post-training approach for multi-hop Retrieval-Augmented Generation (RAG). The method aims to reduce excessive refusals by leveraging the model's internal parametric knowledge when external evidence is missing. It introduces the concept of "fact-level answerability" and utilizes a pipeline involving Supervised Fine-Tuning (SFT) and Direct Preference Optimization (DPO) to generate explainable justifications for refusals. The approach is evaluated on the MuSiQue dataset.

**Reviewer Concerns:**

Since there was no author response, all reviewer concerns remain outstanding. Key concerns include:

Limited Experimental Scope: All reviewers (V1LR, sinM, QfAe) pointed out that evaluating only on the MuSiQue dataset is insufficient to demonstrate generalization.

Unrealistic Evaluation Settings: Reviewers ornH and V1LR noted that the evaluation relied on "clean" contexts without distractors and assumed access to gold-standard question decompositions, which does not reflect real-world RAG scenarios.

Baseline Fairness: Reviewers questioned whether baselines (DTA, TrustAlign) underwent the same SFT stage as the proposed method, casting doubt on the validity of the performance gains.

Presentation and Notation: Reviewers sinM and QfAe found the paper's notation and mathematical formulation difficult to follow and unconventional.

**Reviewer Scores:**

Reviewer Scores

Reviewer ornH (Current: 4): Score would remain unchanged or drop due to lack of clarification on realistic settings.

Reviewer V1LR (Current: 2): Score would remain unchanged as critical flaws in assumptions and baselines were not addressed.

Reviewer sinM (Current: 2): Score would remain unchanged given the complaints about writing quality and lack of reproducibility.

Reviewer QfAe (Current: 4): Score would remain unchanged.

---

### Decision · Program_Chairs · 2026-01-26

Reject